# Winter Road Surface Condition Recognition in Snowy Regions Based on Image-to-Image Translation

**DOI:** 10.3390/s26010241

**Published:** 2025-12-30

**Authors:** Aki Shigesawa, Masahiro Yagi, Sho Takahashi, Toshio Yoshii, Keita Ishii, Xiaoran Hu, Shogo Takedomi, Teppei Mori

**Affiliations:** 1Graduate School of Engineering, Hokkaido University, Sapporo 060-0808, Japan; 2Faculty of Engineering, Hokkaido University, Sapporo 060-0808, Japan; yagi@eng.hokudai.ac.jp (M.Y.); stakahashi@eng.hokudai.ac.jp (S.T.); yoshii@eng.hokudai.ac.jp (T.Y.); 3Bridgestone Corporation, Tokyo 104-8340, Japan; keita.ishii@bridgestone.com (K.I.); xiaoran.hu1@bridgestone.com (X.H.); shogo.takedomi@bridgestone.com (S.T.); teppei.mori@bridgestone.com (T.M.)

**Keywords:** road surface classification, winter road, dusk, image-to-image translation

## Abstract

**Highlights:**

**What are the main findings?**
Illumination conditions are standardized using image-to-image translation, enabling a classification approach that is more robust than time-based switching methods.Illumination normalization via CycleGAN achieved 78% accuracy at dusk, outperforming conventional methods.

**What are the implications of the main findings?**
Enables improved road condition monitoring without relying on unstable time-based model switching.Enhances winter traffic safety by improving the detection of frozen surfaces even under transitional lighting conditions like dusk.

**Abstract:**

In snowy regions, road surface conditions change due to snowfall or ice formation in winter. This can lead to very dangerous situations when driving a car. Therefore, recognizing road surface conditions is important for both drivers and road managers. Road surface classification using in-vehicle cameras faces challenges due to the diverse environments in which vehicles operate. It is difficult to build a single classification model that can handle all conditions. One major challenge is illumination. During dusk, it changes rapidly and drastically, resulting in poor classification accuracy. Therefore, a robust method is needed to accurately recognize road conditions at all times. In this study, we used an image translation method to standardize illumination conditions. Next, we extracted features from both the translated images and the original images using MobileNet. Finally, we integrated these features using Late Fusion with an Extreme Learning Machine to classify road conditions. The effectiveness of this method was verified using a dataset of in-vehicle camera images. The results showed that the accuracy of this method achieved 78% during dusk and outperformed the comparison methods. It was confirmed that the uniformity of illumination conditions contributed to the improvement in classification accuracy. The proposed method can classify road conditions even during dusk, when sudden changes in illumination occur. This demonstrates the potential to realize a robust road condition recognition method that contributes to improved driver safety and efficient road management.

## 1. Introduction

In snowy and cold regions such as Hokkaido, Japan, many traffic accidents occur that are unique to the winter season. The main cause is that vehicle control becomes difficult due to frozen road surfaces and snow accumulation. In such environments, it is desirable to accurately recognize the driving environment. The same is true for automated vehicles, which are expected to be widely used in the future. Furthermore, for administrators and managers in charge of road maintenance and management, it is important to understand the road surface conditions over a wide area for efficient snow removal and the spraying of anti-icing agents.

However, the use of expensive sensors and special equipment for these applications, such as LiDAR or radar, is costly and time-consuming to install and implement in society. In contrast, car-mounted cameras are a more realistic and practical solution, as they are already widely installed in many vehicles. Their widespread use makes them a readily available source for data collection on a societal scale, providing a low-cost and realistic approach to understanding road conditions. Furthermore, modern car-mounted cameras are not only inexpensive but also capable of capturing high-quality images, making them suitable for various vision-based tasks. This allows for the development of software-based systems that can provide useful information to both road users and managers while keeping costs low.

To realize this low-cost approach, various methods have been proposed for recognizing road surface conditions using in-vehicle cameras. Early works relied on handcrafted features [1,2], while recent studies have demonstrated the effectiveness of deep learning approaches [3,4,5,6,7,8,9,10]. However, most of these methods are optimized for distinct illumination conditions, such as clear daytime or nighttime, forcing them to rely on a model switching strategy for use throughout the day. This approach reveals a critical weakness when dealing with dusk, a transitional period during which lighting changes continuously and dramatically. This time of day is particularly important, as it correlates with an increased incidence of traffic accidents [11]. The fundamental challenge is that any model switching strategy based on a time-based threshold is fragile. Previous studies have shown that applying a model trained on a specific illumination domain to a different domain leads to significant performance degradation [12]. A fixed time boundary often fails because sunset times vary significantly by season and location. A dynamic threshold based on daily sunset time is insufficient, as real-world illumination is also affected by weather conditions, artificial lighting, and the surrounding environment. Moreover, the change in lighting during dusk is gradual rather than instantaneous, making it difficult to define a single optimal switching point. Therefore, a robust allday system requires an approach that is not dependent on such time-based triggers.

In this paper, we propose a method that unifies the illumination conditions of images to a consistent reference illumination style defined only for training purposes, rather than a fixed time period using an image-to-image translation and integrates this process into road surface condition recognition. The image translation serves as a preprocessing step, converting images acquired at various times of day into a consistent reference illumination condition. Importantly, this reference style is not used as a temporal boundary during inference; instead, it enables the classifier to operate without knowing the actual time of capture. By reducing the influence of illumination differences, the classifier can focus on learning the intrinsic characteristics of the road surface condition under stable visual conditions. This approach enables more robust road condition classification across all times of day, including dusk, which has been a persistent challenge for conventional methods, thereby contributing to safer driving assistance and more efficient road management.

## 2. Related Works

### 2.1. Road Surface Recognition Using Physical Sensors

The utilization of physical sensors constitutes the most direct approach for the recognition of road surfaces. Gui et al. developed a dual sensor integrating a piezoelectric sensor that utilizes resonance frequency technology and a reflection-type optical sensor [13]. The purpose of this integration is to achieve two objectives: road condition classification and ice/water film thickness measurement. Mori et al. and Ohiro et al. proposed a practical method for directly determining road surface conditions from tire vibrations using CAIS (Contact Area Information Sensing) [14,15]. This method involves installing acceleration sensors inside the tire.

However, these methods require the installation and maintenance of expensive, specialized equipment. This presents cost challenges that hinder widespread adoption in general vehicles and large-scale data collection across society. Conversely, image-based methods that utilize in-vehicle cameras are anticipated to offer a cost-effective and expandable solution, attributable to the increasing prevalence of devices such as dashcams.

### 2.2. Image-Based Recognition with Handcrafted Features

Preliminary studies in the field of image analysis primarily relied on handcrafted features, such as color and texture, as critical elements of analysis. Kawai et al. enhanced robustness by leveraging the blue component in the RGB color space and texture features, thereby accounting for the effects of streetlights and headlights at night [1]. Zhao et al. enhanced the accuracy of recognizing mixed road surfaces by integrating HSV color space moments with GLCM features and leveraging an SVM optimized using PSO [2]. While these methods achieved some success, manually designed features have limitations in representing the diverse road surface variations encountered in complex real-world environments.

### 2.3. Deep Learning-Based Approaches

Recent advancements in convolutional neural networks (CNNs) have dramatically improved the accuracy of road surface condition recognition. Li et al. and Xie and Kwon demonstrated that transfer learning using existing models such as VGG-16 enables efficient learning even with limited data [3,4]. Furthermore, progress has been made in developing lightweight models and constructing large-scale datasets for practical implementation. Ojala and Seppänen proposed the lightweight regression model SIWNet, which achieves high-precision friction coefficient estimation while reducing computational load [5]. Cordes and Broszio, along with Zhao et al., constructed large-scale datasets that encompass diverse road conditions [6,7]. These datasets enhance reliability through uncertainty estimation and decision-level fusion. Furthermore, architectural improvements are advancing. Guo et al. address data imbalance using attention mechanisms, while Krishna and Jyothi integrated DNN features with SVM [8]. Multimodal approaches that leverage non-image information have also been suggested. You and Moroto et al. integrated meteorological and traffic data alongside image features to improve recognition accuracy and enable future predictions [9,10]. Meanwhile, Ishii et al. use spatial machine learning models that consider GIS data and solar radiation to predict high-resolution road surface conditions in urban areas [16].

### 2.4. Generative Approaches for Illumination Normalization

Deep learning-based methods have demonstrated high performance; however, many of these methods assume favorable daytime lighting conditions or are trained specifically for particular lighting environments. Robustness in real-world environments, particularly those with dramatically changing illuminance conditions, remains a significant challenge. To address this domain gap, generative approaches that standardize input styles have gained attention. Wang et al. improved segmentation accuracy in low-light conditions by using CycleGAN to enhance brightness [17]. Similarly, Wu et al. applied GANs to winter road maintenance by translating snow-covered images into clean road images [18]. Romera et al. demonstrated that bridging the day–night gap via style transfer improves classification robustness [19].

Regarding the choice of the generative model, recent state-of-the-art methods such as CUT [20] or MUNIT [21] have shown impressive results in multimodal style transfer. However, these methods may risk altering geometric structures or introducing artifacts in textures, which could be critical for safety-oriented applications. In safety-critical applications like road surface recognition, preserving the texture or physical structure of the road is important. Therefore, this study adopts CycleGAN [22] as the normalization engine. Its cycle-consistency loss explicitly enforces the preservation of structural content while altering only the illumination style, making it a safer choice for this specific task.

### 2.5. Domain Adaptation and Test-Time Strategies

Beyond generative normalization, domain adaptation and test-time adaptation (TTA) are active research areas for handling illumination shifts. Lengyel et al. proposed a zero-shot day–night adaptation method using physics-based priors [23]. More recently, TTA methods have been proposed to adapt the model parameters on the fly during inference to bridge the distribution gap [24]. Self-supervised learning (SSL) also offers a path to learn illumination-invariant features from large-scale unlabeled data.

While these approaches are promising, they typically require iterative model updates, additional optimization steps during inference, or large-scale pretraining with task-specific objectives. Such requirements can introduce instability, increase computational cost, and complicate deployment in real-time or embedded road surface recognition systems.

In contrast, the objective of this study is not to adapt the classifier itself, but to reduce illumination-induced domain gaps through a deterministic preprocessing step that can be applied without modifying the classification model or its training procedure. By performing image-to-image translation as an inference-time normalization, the proposed framework preserves stable and reproducible inference behavior while remaining compatible with lightweight classifiers and real-time constraints.

This design choice allows us to isolate and evaluate the effect of illumination normalization on road surface classification performance, particularly under transitional lighting conditions such as dusk, without introducing additional learning complexity at test-time.

## 3. Road Surface Conditions Recognition Based on Image-to-Image Translation

This chapter describes an image-based method for classifying road surface conditions that takes dusk time into account. Figure 1 shows an overview of the proposed method. In the proposed method, the original image style is transformed and then input into dedicated CNN-based image classification models. The final classification label is obtained by Late Fusion based on features output by four CNN-based image classification models. Section 3.1 describes the individual processing method used in the proposed method, and Section 3.2 describes the specific proposed method.

### 3.1. Individual Processing Used in the Proposed Method

In this section, we describe the individual processes used in the proposed method. In 3.1.1, we describe CycleGAN [22], an image translation method; in Section 3.1.2, MobileNet [25], a lightweight CNN-based image classification model; and in Section 3.1.3, Late Fusion, a data integration method.

#### 3.1.1. CycleGAN

CycleGAN is an image translation method proposed by Zhu et al. [22]. One of its main features is that it does not require datasets that are paired before and after the translation when training the model. The following adversarial loss $L_{GAN}$ is used to learn a mapping G that produces a distribution G(X) that makes the source domain X indistinguishable from the distribution Y of the target domain. Here, x is an image in X and y is an image in Y.(1)$$L_{GAN}\left(G,\;D_{Y},\;X,Y\right)=E_{y\sim p_{data}\left(y\right)}\left[logD_{Y}\left(y\right)\right]+E_{x\sim p_{data}\left(x\right)}\left[log(1-D_{Y}\left(G(x))\right)\right]$$

This causes G to produce a distribution from domain X to domain Y that is as close as possible to the target distribution. However, there are innumerable mappings that can produce such a distribution, and there is also the possibility that they converge to identical outputs. To strengthen the constraint, a cycle-consistency loss $L_{cyc}$ is introduced so that $F(G\left(X\right))$ can recover X using the inverse map F.(2)$$L_{cyc}\left(G,\;F\right)=E_{x\sim p_{data}\left(x\right)}\left[{\left\|F\left(G\left(x\right)\right)-x\right\|}_{1}\right]+E_{y\sim p_{data}\left(y\right)}\left[{\left\|G\left(F\left(y\right)\right)-y\right\|}_{1}\right]$$

This cycle-consistency loss enables training with non-paired images while encouraging the preservation of scene structure and allowing changes in appearance. This property is critical for in-vehicle images, where geometric consistency of road surfaces must be maintained under varying illumination conditions. In practical driving scenarios, it is difficult to acquire paired images across different illumination domains because the vehicle is continuously moving and the surrounding environment changes dynamically. Therefore, CycleGAN was adopted for its suitability in handling non-paired images under such realistic conditions.

#### 3.1.2. MobileNet

MobileNet is a lightweight convolutional neural network architecture proposed by Howard et al. [25], designed specifically for mobile and embedded vision applications. Unlike standard convolutions, MobileNet employs Depthwise Separable Convolutions, which factorize a standard convolution into a depthwise convolution and a 1×1 pointwise convolution. This factorization significantly reduces both the number of parameters and the computational cost while maintaining high recognition accuracy.

In this study, we adopted MobileNet as the feature extractor primarily to align with the baseline method proposed by Kinoshita et al. [12]. By using the same backbone architecture, we ensure that the performance improvements observed in our experiments are attributed to our proposed image translation and fusion strategy, rather than the use of a deeper or more complex feature extractor.

#### 3.1.3. Late Fusion

Late Fusion is a strategy that integrates the outputs of multiple independent classifiers to make a final decision [26]. In the context of this study, distinct illumination conditions possess unique feature distributions. Therefore, rather than relying on a single monolithic model, it is more effective to employ specialized models for each domain and aggregate their judgments. This approach leverages the complementary strengths of each classifier, leading to more robust recognition [27].

For the aggregation method, we employ an Extreme Learning Machine (ELM) [28]. Since the input to the fusion layer consists of low-dimensional probability vectors (specifically, 24 dimensions as described in Equation (4)), employing complex deep neural networks would be prone to overfitting and would entail unnecessary computational costs. ELM, which is a single-hidden-layer feedforward network with randomized input weights, offers extremely fast learning speeds and high generalization capability, making it ideal for this lightweight fusion task.

### 3.2. Road Surface Condition Recognition Method Considering Dusk Time

This section describes the specific flow of the image-based road surface condition classified method considering dusk time. The following sections explain the image translation using CycleGAN in Section 3.2.1, feature calculation using MobileNet in Section 3.2.2, and Late Fusion using ELM in Section 3.2.3.

#### 3.2.1. Image Translation Using CycleGAN

In this section, we explain how the CycleGAN image translation, which is the key to the proposed method, is performed. As discussed in previous sections, continuously changing illuminance acts as noise for road surface classification, and relying on a time-based trigger to switch models is not a robust solution. To overcome this, our study proposes an approach that standardizes the illumination conditions of images before they are fed to the CNN-based image classifier. Specifically, we define two target stylistic domains: “typical daytime (Core-Day)” and “typical nighttime (Core-Night)”, as shown in Figure 2. Core-Day and Core-Night are defined not as fixed time periods, but as periods during which fluctuations in illuminance conditions are small and typical states of that domain persist stably. In other words, they serve as “style templates” for the image translation model. During the training phase, we use a collection of images captured under stable day and night conditions to teach CycleGAN what these typical styles look like. The goal is to train generators that can convert the style of any input image to match these learned Core-Day and Core-Night appearances, regardless of the actual time the image was captured. This allows the image classifier to always classify road surface conditions under standardized illuminance conditions, and it is expected that the classifier will be able to focus mainly on differences in road surface conditions.

#### 3.2.2. Feature Extraction Through MobileNet

In the proposed method, four MobileNet models are trained and used individually according to illumination conditions and image translation styles. Specifically, the input stream is designed to handle uncertainty in the real-world environment. Since the system does not rely on a clock or GPS to determine whether it is day or night, the original (non-translated) image is fed into both the Day-trained and Night-trained MobileNets in parallel. By consulting both “experts”, the system ensures that relevant features are captured regardless of the actual time of day. Each model is trained using the image types shown in Table 1. They are trained to output the probability (confidence score) of belonging to each road surface condition class, using a SoftMax function in the last fully connected layer. This results in four different viewpoint-based confidence scores p from a single image, as shown in Equation (3). Here, k is one of the road surface conditions and t is an image type.(3)$$p_{t}^{k}\left(\begin{array}{c} k\in \left\{Dry,\;Semiwet,\;Wet,\;Slush,\;Ice,\;Snow\right\}\\ \;t\in \left\{\begin{array}{c} Day,Night,\;\\ Generated-Core-Day,\\ Generated-Core-Night \end{array}\right\} \end{array}\right)$$

#### 3.2.3. Late Fusion by ELM

The outputs from the four MobileNets obtained in the previous section are combined to determine the most probable road surface condition. Specifically, the confidence of each model obtained in the previous section for each class is combined into a 24-dimensional feature vector $f_{image}$, as shown in Equation (4). This allows us to obtain the final label for a single image by comprehensively considering the confidence from all four models. Here, GC denotes Generated Core.(4)$$\begin{array}{c} f_{image}=\left\{p_{Day}^{k},\;p_{Night}^{k},\;p_{GCDay}^{k},\;p_{GCNight}^{k}\right\}\\ \left(k\in \left\{Dry,\;Semiwet,\;Wet,\;Slush,\;Ice,\;Snow\right\}\right) \end{array}$$

Thus, by integrating the knowledge of multiple classifiers with ELM, it is expected to improve the robustness and overall decision reliability in situations where any single viewpoint alone would be incorrect.

## 4. Experiments

In this chapter, we conduct experiments to verify the effectiveness of the proposed method. Section 4.1 describes the in-vehicle camera dataset used in the experiments, Section 4.2 presents the experimental setup, and Section 4.3 reports the experimental results.

### 4.1. Description of the Onboard-Camera Dataset

Figure 3 shows examples of the in-vehicle camera images used in the experiments. The images were taken during the time of 12:00~20:00 in the suburbs of Sapporo, Hokkaido, Japan, between December 2022 and March 2023. The image resolution is 800 × 600 pixels. The driving routes include mountainous areas, urban roads, and expressways, as shown in Figure 4. By incorporating diverse road conditions, lighting environments, and background scenes, the dataset reflects the complexity of real driving environments in winter.

The dataset includes the date and time of the shooting, latitude and longitude data acquired by GPS sensors, and road surface condition labels recorded by an experienced annotator during the shooting. Since the images were taken continuously while the vehicle was in motion, images that were judged to have high similarity using imgsim [29], as well as images in which the road surface was significantly occluded by other objects such as preceding vehicles or buildings, were removed. After this filtering process, we compiled a balanced dataset containing about 3500 images per class for the daytime period (12:00–16:00) and about 3500 images per class for the nighttime period (16:00–20:00), ensuring sufficient data volume for both training and testing.

### 4.2. Experimental Settings

The road surface condition labels are Dry, Semi-Wet, Wet, Slush, Ice, and Snow. The dataset was divided into training, validation, and test sets as follows. For evaluation, a separate test dataset was prepared, consisting of 100 images per class for each time period (Core-Day, Core-Night, and Dusk), except for the Ice class during dusk. These test images were strictly separated from the training and validation sets to ensure fair evaluation. The remaining images were used for training and validation, as summarized in Table 2. The data split was performed randomly using a fixed random seed to ensure repeatability of the experiments.

For CycleGAN training, input images were resized to 286 × 286 pixels and randomly cropped to 256 × 256 pixels. During inference, no preprocessing was applied, and the output resolution was 800 × 600 pixels. Each domain contained 1000 images per class, resulting in 6000 images in total. The model was trained for 200 epochs, with a learning rate of 2 × 10^−4^ for the first 100 epochs and linearly decayed to zero over the remaining epochs. The batch size was set to 1.

MobileNet is based on the MobileNetV3-Large model pre-trained on ImageNet [30], and all layers were fine-tuned. During training, we pre-processed the images by resizing them to 224 × 224 pixels, random horizontal flipping and rotation (±10°), tonal adjustment, and normalization based on the ImageNet mean and standard deviation (mean: [0.485, 0.456, 0.406], std: [0.229, 0.224, 0.225]). Optimization was performed using Adam [31], and the learning rate was set to $1\times 10^{-4}$. To prevent overfitting, training was terminated if the validation loss did not improve for five consecutive epochs during training. For Day and Night models, each class has 2000 images, of which 80% of them were used for training and the remaining 20% were used for validation during training. For Generated-Core-Day and Generated-Core-Night models, each class has 4000 images, which were separated into training and validation sets using the same ratio.

For ELM, the size of the input layer, the number of neurons in the hidden layer, and the size of the output layer were set to 24, 350, and 6, respectively. ReLU was used as the activation function of the hidden layer, and the weights from the input layer to the hidden layer and the bias of the hidden layer were determined by He initialization. The regularization parameter λ in the output layer weight calculation was set to $1\times 10^{-6}$. These hyperparameters, including the hidden layer size and the regularization parameter, were determined empirically through preliminary experiments. Each class has 500 images from the Day and Night domains, respectively.

All deep learning models, including CycleGAN and MobileNet, were implemented using the Pytorch framework (version 2.6.0, Meta Platforms, Inc., Menlo Park, CA, USA). The ELM was implemented using the scikit-learn library. All training and inference processes were conducted on a workstation equipped with an NVIDIA GeForce RTX 4080 GPU (NVIDIA Corp., Santa Clara, CA, USA) and an Intel Core i5-9400 CPU (Intel Corp., Santa Clara, CA, USA).

The comparative methods are illustrated in Figure 5. The first (CM1) is a method proposed in [12] that uses Late Fusion of Day and Night classifiers. The second (CM2) is a method that trains MobileNet by mixing Day and Night images and performs classification. The third method is to input images into MobileNet trained on only Day images (CM3-Day) or only on Night images (CM3-Night) for classification.

The F-Score and accuracy shown in the following equation are used to compare the methods. Here, TP stands for True Positive, FP for False Positive, TN for True Negative, and FN for False Negative.(5)$$\begin{array}{c} F\text{-}Score=\frac{2\times Precision\times Recall}{Precision+Recall}\\ Precision=\frac{TP}{TP+FP},\;Recall=\frac{TP}{TP+FN}\\ \;Accuracy=\frac{TP+TN}{TP+TN+FP+FN} \end{array}$$

For the sake of convenience, the results of this experiment are divided into time periods, as shown in Table 3. Note that these time period definitions are specific to the winter season in Hokkaido, Japan, and may vary in other regions or seasons due to differences in sunset times. To determine the dusk period, the sunset time was calculated using Astral libraries [32] based on date, latitude, and longitude data as inputs. It is important to note that these time ranges were used solely to construct stable reference domains for training and to organize evaluation datasets. The proposed method itself does not require or use any time information during inference.

### 4.3. Experimental Results and Discussion

#### 4.3.1. Quantitative Analysis

Table 4 shows the experimental results for the Core-Day and Core-Night test data. The proposed method (PM) achieved an overall accuracy of 0.80 in both Core-Day and Core-Night conditions, outperforming comparison methods CM1–CM3 in most classes. CM3 demonstrates performance comparable to the proposed method when handling the same domain during training and testing. However, accuracy drops dramatically when nighttime images are input into the daytime model or vice versa. For example, in summer or in more southerly regions, the sunset time is delayed significantly, and the 16:00 threshold value does not adequately separate Day and Night, resulting in a situation where the Day model processes Night images and the Night model processes Day images. In contrast, our method provides a robust solution by standardizing the image style, making it independent of these fragile temporal triggers. Consequently, the superiority of the proposed method, which enables stable, high-accuracy classification without requiring model switching, is confirmed.

Table 5 shows the experimental results for dusk test data. The dusk period has been a persistent challenge in previous studies due to drastic changes in illumination conditions. The proposed method showed a notable improvement with accuracy of 0.78, whereas CM1 dropped to 0.63 and CM3 dropped to 0.55 and 0.44 for the Day and Night modes, respectively. In this case, CM2 recorded the second-highest accuracy. Since CM2 is trained using both day and night images, it may have a better ability to extract features from dusk images, which are in between day and night.

To verify the statistical reliability of the experimental results, we conducted McNemar’s test with Bonferroni correction (α=0.005). The analysis was performed for the entire test dataset (overall) as well as for each time period (Day, Night, and Dusk) individually. Table 6 summarizes the results of the statistical tests comparing the proposed method with other methods.

In the overall evaluation across all time periods, PM showed a statistically significant difference (p<0.001) against all comparison methods. This suggests that the proposed method is highly robust throughout the day.

In the challenging dusk period, although there was no statistically significant difference in terms of overall accuracy between PM and CM2 (p>0.05), a clear distinction emerges when focusing on safety-critical performance. As shown in Table 5, PM achieved a higher F-Score of 0.67 for the Ice class compared to 0.48 for CM2. This disparity is particularly vital because detecting frozen surfaces is more critical for winter driving safety than identifying other conditions. Therefore, while both methods are statistically comparable in aggregate, the proposed method provides a substantial practical advantage in real-world, high-risk scenarios.

In addition, confusion matrices for the results of PM, CM1, CM2, CM3-Day, and CM3-Night are shown in Figure 6, Figure 7, Figure 8, Figure 9 and Figure 10. Figure 6, which presents the confusion matrices for the proposed method, reveals that the Semi-Wet and Wet classes remain the most challenging classes. In contrast, Dry and Snow surfaces are consistently recognized with high-accuracy, even under dusk conditions. Furthermore, the ‘Allday’ matrices in Figure 6, Figure 7, Figure 8, Figure 9 and Figure 10 demonstrate that the proposed method maintains stable and high performance across the entire dataset. The number of misclassifications increases in Ice and Snow in CM1-CM3, whereas PM shows fewer misclassifications. Figure 7 shows that CM1 exhibits a tendency similar to PM during Day and Night; however, misclassifications increase during Dusk, confirming its vulnerability. Figure 8 confirms that CM2 is similar to PM during Dusk but exhibits higher error rates during both Day and Night. While CM2 possesses the ability to extract features from both daytime and nighttime, it may be inferior to PM or CM1, which utilize models specifically trained for each scenario. Figure 9 and Figure 10 confirm that classification performance significantly deteriorates when the training and testing domains differ. This is particularly noticeable in classes like Dry, where many correct predictions are made by other methods but are misclassified. Furthermore, Dusk also exhibits a high number of misclassifications. These findings indicate that CM3 does not function effectively unless model switching based on perfect threshold settings is possible, which is often difficult to achieve.

#### 4.3.2. Qualitative Analysis

Figure 11, Figure 12 and Figure 13 show examples of images that were correctly recognized by the proposed method. In each figure, the first row shows the original input images. The second and third rows display the images translated into Core-Day and Core-Night styles, respectively. Figure 11 and Figure 12 show the results for Core-Day and Core-Night inputs. In these cases, the generator acts as an identity mapping or performs a natural style transfer (e.g., turning a day scene into a realistic night scene), confirming that the model behaves stably even when the input is already within a standard domain.

More importantly, Figure 13 visualizes the image translation results specifically for the challenging “Dusk” condition. As seen in the first row of Figure 13, the original input images are distributed across diverse illumination conditions. However, the CycleGAN-based translation successfully maps these diverse inputs into uniform illumination conditions that closely resemble the standard “Core-Day” and “Core-Night” domains.

The quantitative results of CM3 in Section 4.3.1 have already demonstrated that CNN classifiers perform optimally when the illumination conditions of the input data match those of the training data. By converting ambiguous or transitional lighting into these consistent styles, the proposed method effectively aligns the input distribution with the training distribution. This allows domain-specific CNNs to operate with high confidence and accuracy, mirroring the ideal conditions observed in CM3.

Although PM demonstrated superior performance compared to existing methods, challenges remain. The core issue lies in image translation, which is the key component of the proposed method. CycleGAN was originally designed solely for visual style transfer. While this paper employs it for its potential contribution to illuminance condition uniformization, the transformation process risks losing image features useful for road surface condition recognition or generating noise. Figure 14, Figure 15 and Figure 16 show an example in which the proposed method failed to correctly recognize the road surface condition. In each figure, the original image is shown on the left, and the translated image is shown on the right. In Figure 14, the correct label is Wet. However, when the image is translated into the Core-Day style, a white texture appears on the road surface. The proposed method recognizes this image as snow, and the occurrence of this texture may have influenced that determination. In Figure 15, the correct label is Semi-Wet, but the proposed method recognized the image as Wet. In this example, reflections on the road surface may have been rendered more prominently than in the original image. Finally, in Figure 16, the correct label is Snow, but the proposed method recognized it as Ice. In this case, the snow-covered road surface, which is nearly pure white, appears grayish after image translation. This resembles a state in which snow has been compacted by vehicle traffic and frozen into an icy surface.

These examples suggest that current image translation methods may not be optimal for road surface condition recognition. These failure cases highlight the inherent domain adaptation challenge between translated images and real samples. While image translation reduces illumination discrepancies, it may also introduce appearance shifts that partially distort discriminative surface cues. While misclassifying snow as ice (as seen in Figure 16) can be considered a conservative error from a safety standpoint, the opposite case—misclassifying ice as snow—poses a significantly higher risk to traffic safety by underestimating road slipperiness. This limitation indicates that illumination normalization alone is insufficient, and future work should incorporate structure-preserving or uncertainty-aware adaptation strategies. Exploring more suitable methods for uniformizing illumination conditions could potentially improve recognition performance.

#### 4.3.3. Computational Efficiency

To assess the feasibility of deploying the proposed method on in-vehicle systems, we evaluated the model size and inference speed. The proposed system consists of two CycleGAN generators, four MobileNetV3-Large, and a single ELM. The file size of a single CycleGAN generator is approximately 45 MB, a single MobileNet v3-large is approximately 17 MB, and a single ELM is approximately 85 KB. Consequently, the total storage requirement for the entire system is approximately 158 MB. This size is sufficiently compact for deployment on modern edge computing devices.

Regarding inference speed, the inclusion of CycleGAN is the primary computational load. In our experimental environment (NVIDIA GeForce RTX 4080), the processing time for CycleGAN image translation was approximately 37.1 ms per image, while the processing time for MobileNetV3-Large was approximately 5.1 ms per image. It should be noted that image translation using CycleGAN can potentially be applied at a lower frequency in practical driving scenarios. Since illumination conditions change gradually, translated reference images can be updated at a lower frequency. Therefore, the overall computational burden can be significantly reduced in real-time operation, and the proposed framework is feasible for in-vehicle systems.

## 5. Conclusions

This paper proposes a method to normalize the illuminance conditions by introducing image translation using CycleGAN to improve classification of road surface conditions using in-vehicle camera images. The proposed method transforms images acquired at any time of day into domains of standard illuminance conditions, namely Core-Day and Core-Night, using CycleGAN, and performs Late Fusion of features obtained from both the transformed images and the original images using ELM. This enables improved road surface classification in winter driving environments with changing illumination conditions. In the experiments, we evaluated the proposed method and several comparison methods using in-vehicle camera images collected in the suburbs of Sapporo, Hokkaido, Japan, during the winter season. The results confirmed that the proposed method has a classification accuracy of 78% even during dusk, when illumination conditions are particularly challenging. However, limitations of the current image translation method were also identified. Therefore, further investigation into more suitable methods of uniformizing illumination conditions is required. These findings are expected to support future research toward improving driver safety and efficient road management by enabling accurate road condition classification throughout the day, including dusk, without relying on time-based boundaries.

## Figures and Tables

**Figure 1 sensors-26-00241-f001:**
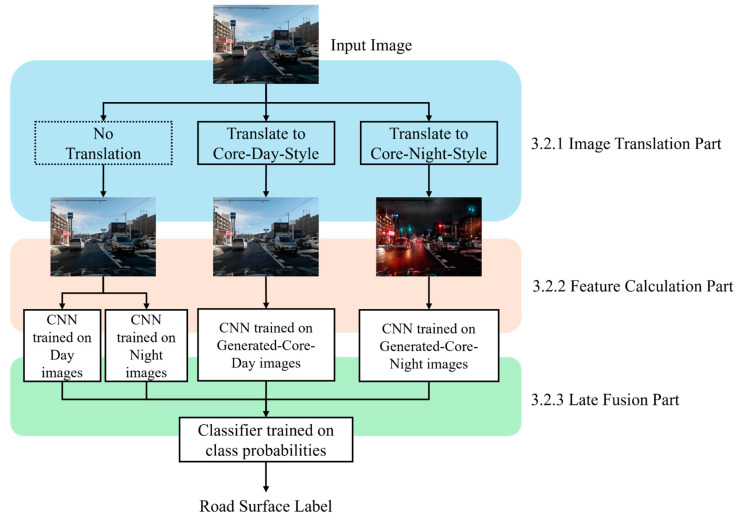
Overview of the proposed method. The input image is processed through three branches: one without translation, one translated to the Core-Day style, and one translated to the Core-Night style. In the feature calculation stage, the original (non-translated) image is fed into two separate CNNs trained on day and night images, respectively. Meanwhile, the translated images are processed by their corresponding style-specific CNNs. Finally, the output probabilities from these four CNNs are concatenated and fed into a classifier for Late Fusion to predict the road surface label.

**Figure 2 sensors-26-00241-f002:**
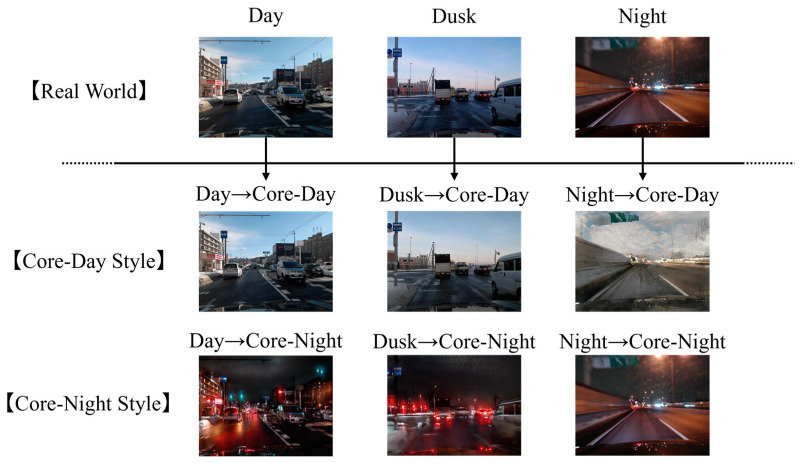
Visualization of image translation into Core-Day and Core-Night styles. Input images captured at various times (Day, Dusk, and Night) are unconditionally translated into both the Core-Day and Core-Night domains. This process standardizes the illumination conditions, ensuring that ambiguous “Dusk” images are mapped to stable feature spaces regardless of the actual recording time.

**Figure 3 sensors-26-00241-f003:**
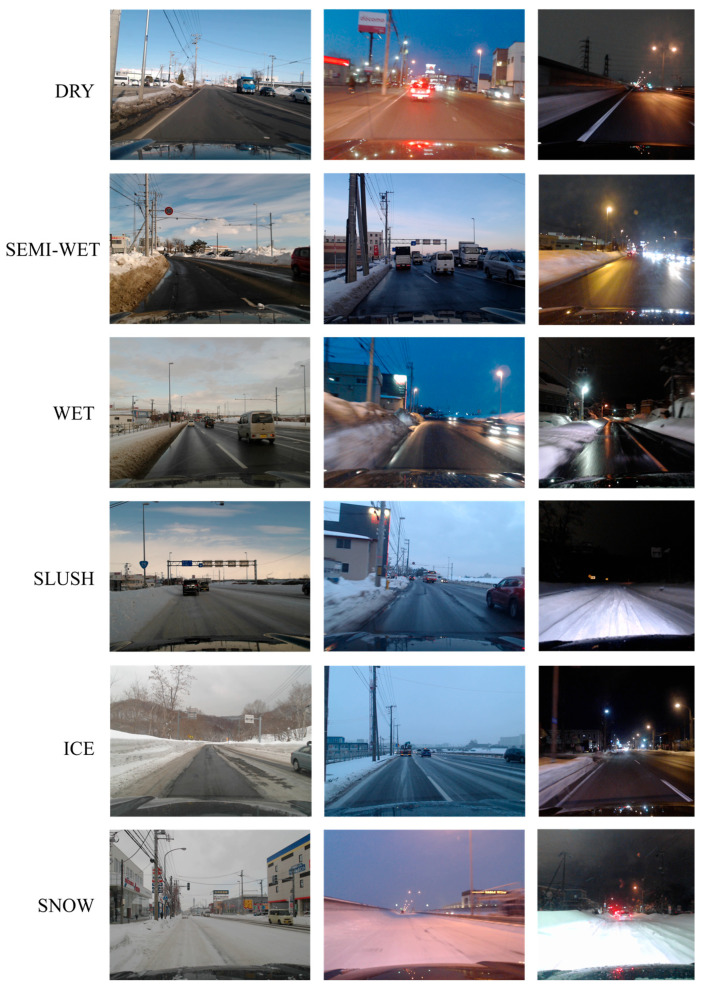
Examples of in-vehicle camera images used in the experiment.

**Figure 4 sensors-26-00241-f004:**
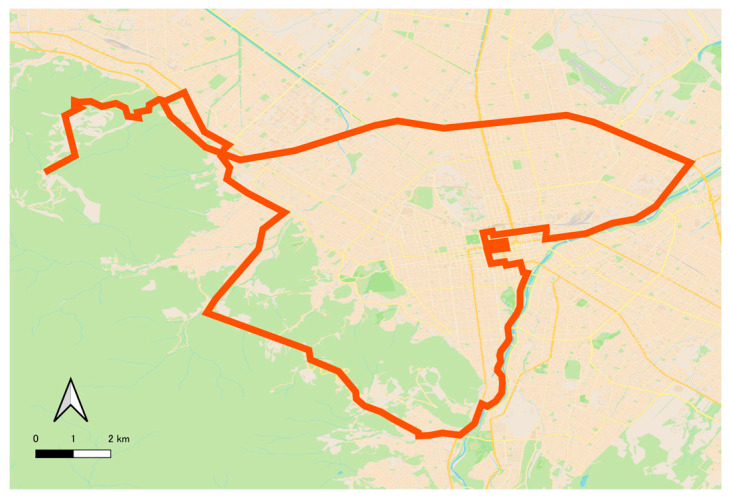
Driving route for data collection. The data was collected in the city of Sapporo, Hokkaido, Japan. The route includes mountainous areas, urban areas, and expressways.

**Figure 5 sensors-26-00241-f005:**
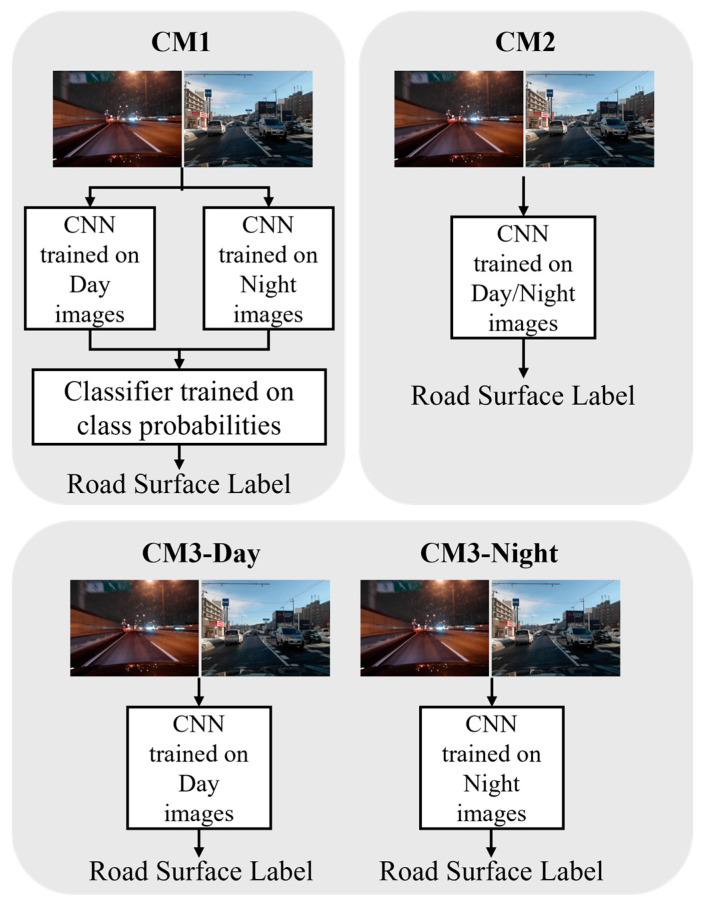
Overview of the comparative method ((**Top Left**): CM1, (**Top Right**): CM2, and (**Bottom**): CM3).

**Figure 6 sensors-26-00241-f006:**
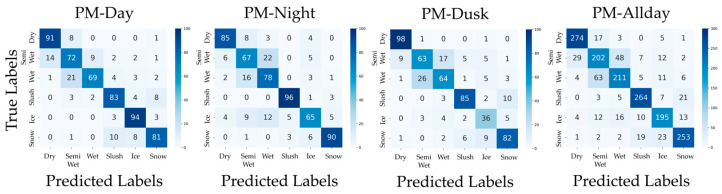
Confusion matrices for PM (Day, Night, Dusk, and Allday).

**Figure 7 sensors-26-00241-f007:**
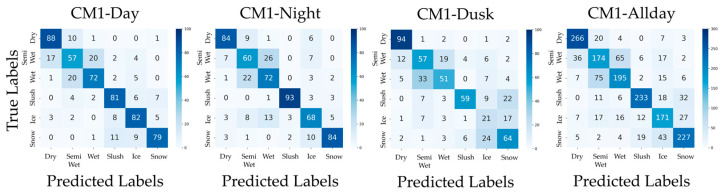
Confusion matrices for CM1 (Day, Night, Dusk, and Allday).

**Figure 8 sensors-26-00241-f008:**
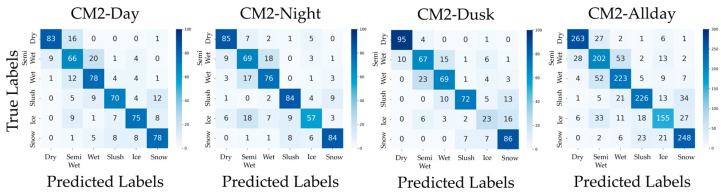
Confusion matrices for and CM2 (Day, Night, Dusk, and Allday).

**Figure 9 sensors-26-00241-f009:**
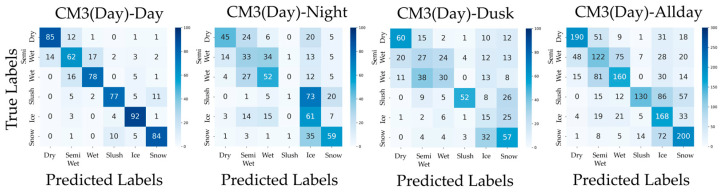
Confusion matrices for CM3-Day (Day, Night, Dusk, and Allday).

**Figure 10 sensors-26-00241-f010:**
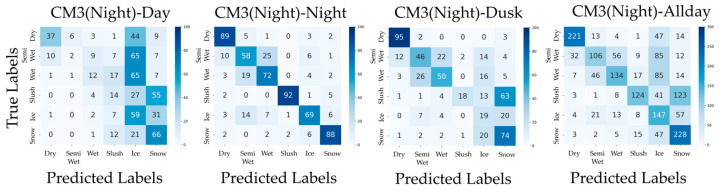
Confusion matrices for CM3-Night (Day, Night, Dusk, and Allday).

**Figure 11 sensors-26-00241-f011:**
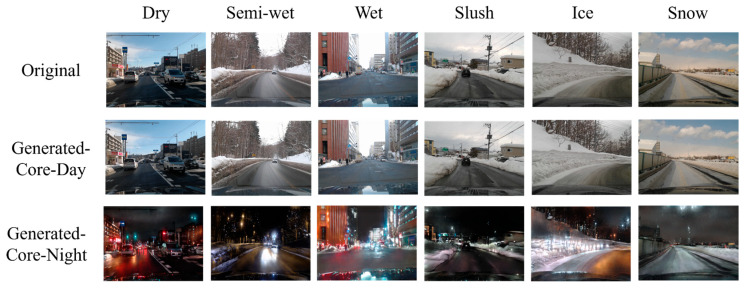
Examples where the proposed method was effective in Core-Day.

**Figure 12 sensors-26-00241-f012:**
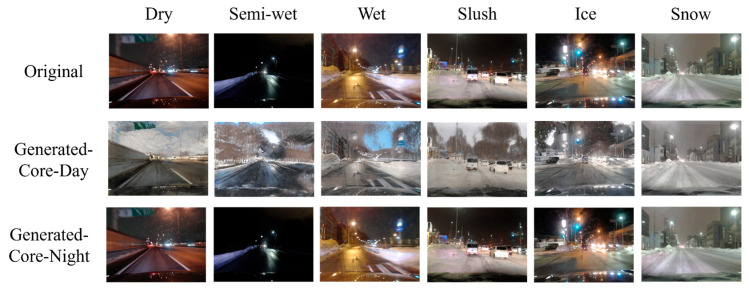
Examples where the proposed method was effective in Core-Night.

**Figure 13 sensors-26-00241-f013:**
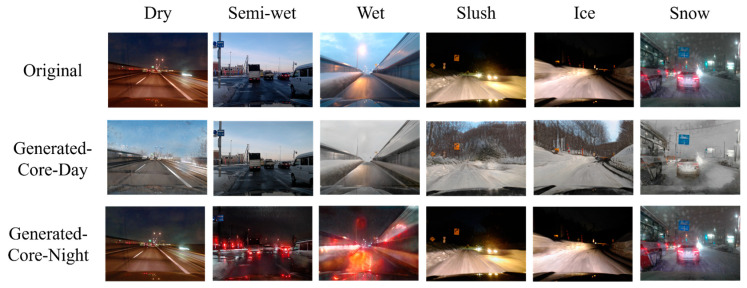
Examples where the proposed method was effective in Dusk.

**Figure 14 sensors-26-00241-f014:**
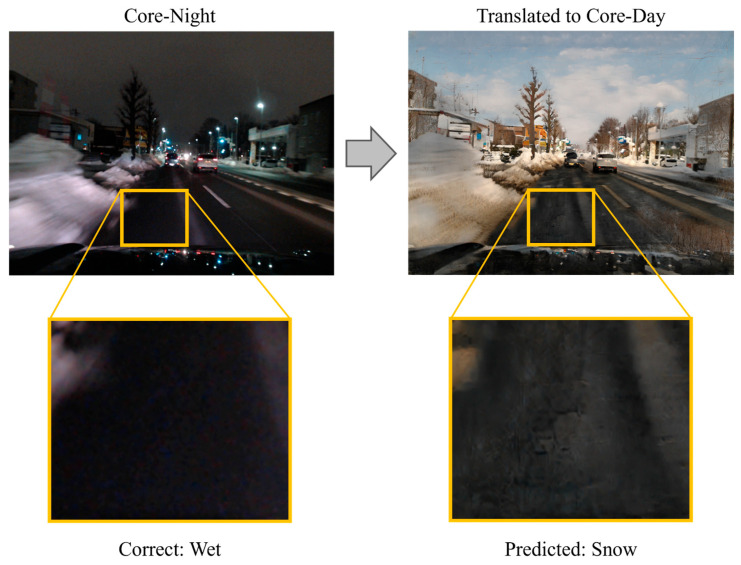
Example where the proposed method recognized Wet surface as Snow surface.

**Figure 15 sensors-26-00241-f015:**
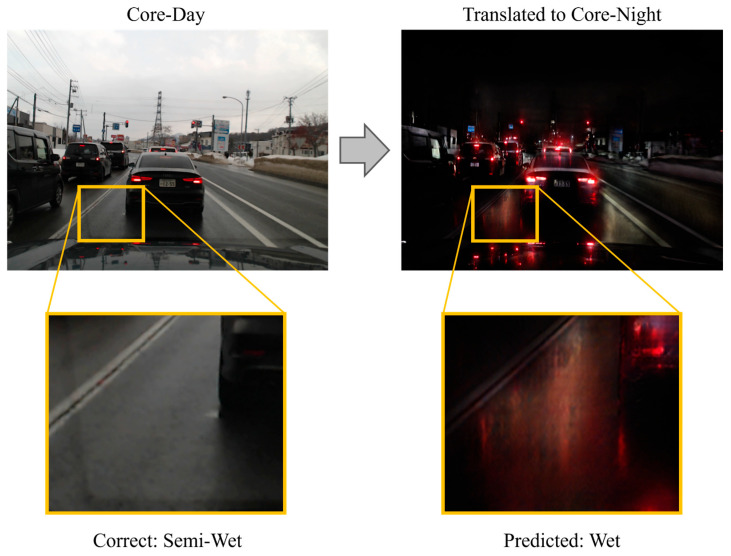
Example where the proposed method recognized Semi-Wet surface as Wet surface.

**Figure 16 sensors-26-00241-f016:**
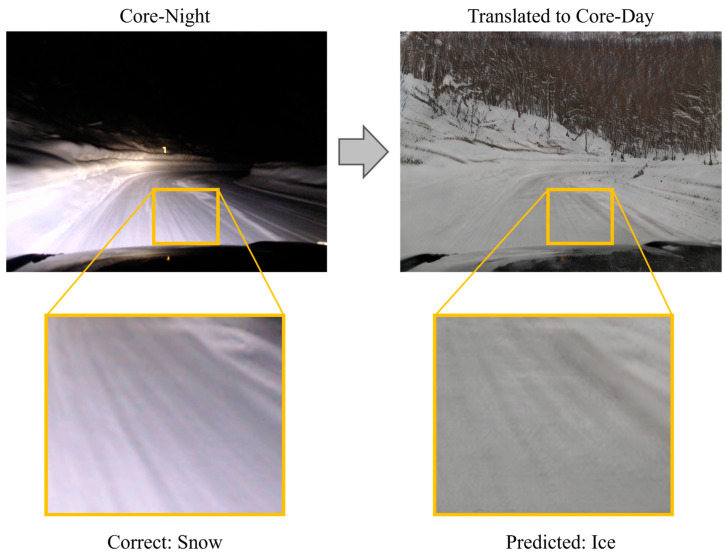
Example where the proposed method recognized Snow surface as Ice surface.

**Table 1 sensors-26-00241-t001:** Image types and their explanations used to train four MobileNets.

Image Type	Explanations
Day	Daytime Images
Night	Nighttime Images
Generated-Core-Day	Images Translated into Core-Day Style
Generated-Core-Night	Images Translated into Core-Night Style

**Table 2 sensors-26-00241-t002:** Number of training data images for one class.

Model to Train	Training Data Images
CycleGAN	1000
MobileNet for Day/Night	2000
MobileNet for Generated-Core-Day/Night	4000
Extreme Learning Machine	500

**Table 3 sensors-26-00241-t003:** Time periods divided in the experiments.

Time Periods	Time
Day	12:00~16:00
Night	16:00~20:00
Dusk	1 h before or after sunset time
Core-Day	12:00~16:00 and not dusk
Core-Night	16:00~20:00 and not dusk

**Table 4 sensors-26-00241-t004:** F-Score and accuracy of each class for Core-Day and Core-Night (bold for the best, underlined for the second-best).

		Core-Day					Core-Night					
		PM	CM1	CM2	CM3-Day	CM3-Night	PM	CM1	CM2	CM3-Day	CM3-Night	Test
F-Score	Dry	**0.88**	0.84	0.86	0.85	0.50	**0.86**	0.85	0.83	0.54	0.85	100
	Semiwet	**0.70**	0.59	0.63	0.62	0.04	**0.67**	0.60	0.65	0.33	0.59	100
	Wet	0.77	0.73	0.73	**0.79**	0.18	0.73	0.68	**0.74**	0.49	0.70	100
	Slush	**0.82**	0.79	0.74	0.80	0.18	**0.94**	**0.94**	0.83	0.02	**0.94**	100
	Ice	**0.89**	0.80	0.77	0.87	0.32	0.71	0.69	0.65	0.39	**0.73**	100
	Snow	0.83	0.82	0.78	**0.84**	0.47	**0.90**	0.87	0.84	0.59	0.86	100
	Accuracy	**0.82**	0.77	0.75	0.80	0.32	**0.80**	0.77	0.76	0.42	0.78	600

**Table 5 sensors-26-00241-t005:** F-Score and accuracy of each class for dusk (bold for the best, underlined for the second-best).

		Dusk					
		PM	CM1	CM2	CM3-Day	CM3-Night	Test
F-Score	Dry	**0.94**	0.88	0.93	0.90	0.62	100
	Semiwet	0.65	0.55	**0.67**	0.50	0.28	100
	Wet	0.67	0.56	**0.70**	0.55	0.35	100
	Slush	**0.85**	0.69	0.79	0.30	0.65	100
	Ice	**0.67**	0.36	0.48	0.29	0.21	50
	Snow	**0.82**	0.61	0.79	0.55	0.47	100
	Accuracy	**0.78**	0.63	0.75	0.55	0.44	600

**Table 6 sensors-26-00241-t006:** Results of McNemar’s test between the proposed method and compared methods. (○: significant difference (p<0.005), ×: no significant difference).

Comparison Pair	Core-Day	Core-Night	Dusk	Overall
PM vs. CM1	×	×	○ (*p* < 0.001)	○ (*p* < 0.001)
PM vs. CM2	○ (*p* < 0.001)	×	×	○ (*p* < 0.001)
PM vs. CM3-Day	×	○ (*p* < 0.001)	○ (*p* < 0.001)	○ (*p* < 0.001)
PM vs. CM3-Night	○ (*p* < 0.001)	×	○ (*p* < 0.001)	○ (*p* < 0.001)

## Data Availability

The datasets presented in this article are not readily available because of the confidentiality agreements with the data provider.

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
