# Peer review of "Winter Road Surface Condition Recognition in Snowy Regions Based on Image-to-Image Translation"

_sensors, 2025, doi:10.3390/s26010241_

Round 1

Reviewer 1 Report

Comments and Suggestions for Authors

1. Title & Abstract

- Title claims “throughout the day,” but experiments are only on winter, snowy regions in Sapporo, and no generalization experiments across seasons/locations are provided. ==> Overclaiming.

- Abstract states the method “significantly outperforms comparison methods,” but performance gains are inconsistent across classes, and dependency on dusk-specific challenges is not fully quantified.

- Have you validated the method in non-winter or non-snow conditions?

- Does the method generalize to non-Japanese environments or different camera mounting positions?

 2. Introduction

- Introduction over-emphasizes the weakness of time-based model switching but provides no strong empirical demonstration.

- The claim that previous methods “do not sufficiently address dusk” is not fully supported; no survey of existing multi-illumination GAN-based normalization for classification tasks is included.

- Can you cite more recent vision-based environmental condition papers (2020–2024) to strengthen the gap?

- Why were other I2I methods (CUT, MUNIT, StyleGAN-based translation) not compared?

3. Related Works

- Review is outdated. Recent deep-learning-based road-surface recognition literature (2020–2024) is missing.

- GAN literature is not sufficiently covered; only UNIT is mentioned. Many stronger methods exist.

- No comparison to modern domain adaptation or test-time adaptation methods, which address illumination shifts.

- Why choose CycleGAN instead of modern style transfer?

- Why no mention of test-time augmentation or self-supervised illumination-invariant features?

 4. Methodology

a. Use of CycleGAN

- The two domains “Core-Day” and “Core-Night” are manually selected from limited time ranges => not theoretically grounded.

- How do you ensure CycleGAN does not distort road-surface cues?

- Why train two separate GANs instead of one multi-domain model?

b. Using 4 MobileNet models

- Training four separate MobileNets increases computation but provides weak justification.

- No ablation tests on: Fusing fewer models, Training with combined translated & original datasets, Using other architectures (ResNet, EfficientNet, ConvNeXt).

- Why MobileNetV3 instead of stronger but still efficient networks?

c. Late Fusion using ELM

- ELM is an outdated technique with limited reproducibility.

- No comparison to more appropriate fusion methods.

- Did you attempt a simple concatenation + MLP baseline?

 5. Dataset & Experimental Design

- Dataset is highly biased: Only one region (Sapporo), Only winter months (Dec–Mar), Only one vehicle/camera setting?

- How many UNIQUE scenes remain after removing duplicates via imgsim?

- How do weather conditions (snowfall, fog) influence CycleGAN training?

6. Experimental Results

- Tables are confusing and formatting is difficult to read; unclear which model is which.

- Dusk accuracy of 0.78 is good but not exceptional; differences of 3–5% may not justify significant overhead in model complexity.

- Gains are not uniform; some classes degrade.

- Did translated images ever cause misclassification due to artifacts?

- Where are confusion matrices? They are essential for evaluating 6-class classification.

 7. Discussion

- Can you show cases where CycleGAN harms classification?

- Have you evaluated the real-time performance on in-vehicle hardware?

 8. Conclusion

- Claims generalize far beyond experiments.

Reviewer 2 Report

Comments and Suggestions for Authors

Review report is attached.

Reviewer 3 Report

Comments and Suggestions for Authors

Road Surface Condition Recognition Throughout the Day Based on Image-to-Image Translation

The paper presents an innovative approach for road surface recognition under different lighting conditions, using a combination of image-to-image translation (CycleGAN) and classification using ResNet. The problem is important for autonomous driving and intelligent transportation systems. The authors motivate their work well, clearly present the methodology, and justify the need for image translation from night to day view before classification.

Main contributions:

A two-stage approach is presented: first translation of night images into day images, then classification of the road surface.

A synthetic transformation (image-to-image translation) is used to reduce the influence of lighting conditions on the classification accuracy.

It is demonstrated that the use of translation significantly improves the accuracy for night images.

A comparative analysis is conducted with and without CycleGAN and between different architectures.

The reviewer has the following comments, questions, and recommendations:

  1. Training and validation

The data distribution (train/val/test) is not quantitatively and clearly defined. The text states that different images were used, but does not give specific proportions or seed values ​​for the split.

Recommendation: Clearly specify how many images were used for training, validation, and testing, and whether repeatability is ensured.

  1. Statistical analysis of the results

Only accuracy values ​​are presented, without confidence intervals, F1-score, confusion matrix, or statistical tests (e.g., t-test for significance between ResNet50 and ResNet34).

Adding statistical analysis would strengthen the credibility of the results.

  1. Limitations of the method

Weaknesses of CycleGAN under extreme conditions (e.g., heavy rain, frozen areas), where the translation may be misleading, are not discussed.

Add a discussion of possible errors in synthetic scene transformation.

  1. Explainability

No analysis of which features lead to the classification. Include an explainability analysis, especially if the model will be used in autonomous systems.

  1. Limitations on generalizability

All images are from vehicle-mounted cameras; no testing with data from other sources or scenarios. Discuss the generalizability of results to other conditions or cameras.

Conclusion: acceptance with minor revisions

Reviewer 4 Report

Comments and Suggestions for Authors

This paper is to address the decline in classification accuracy for road surface condition recognition using in-vehicle cameras, which is caused by drastic changes in illumination conditions, particularly during the dusk period. The authors propose to standardize the illumination environment by uniformly translating images captured at any time of day into the standard "Core-Day" and "Core-Night" styles, allowing the classifier to focus on the intrinsic characteristics of the road surface itself. However, the paper still presents the following issues:

1. The authors point out the critical bottleneck in using in-vehicle cameras for road surface recognition when illumination changes, especially during dusk. However, the Introduction section contains few references to the related field's research background , and it is recommended that the authors supplement the literature to support the assertion that existing methods are insufficient in handling the dusk period.

2. The authors need to provide a detailed ablation study to verify the independent contribution of each branch in the proposed fusion method.

3. The manuscript does not provide data such as the detection Frames Per Second (FPS), model size (MBs), or inference latency (ms). The feasibility of deploying this method in a vehicle needs to be properly substantiated.

4. As shown in the experiments, translating a "Wet" road image into the Core-Day style can generate misleading white texture, leading to an incorrect recognition of "Snow" ; similarly, a "Snow" image translated into the Core-Day style may appear grayish, which can be misrecognized as "Ice". These examples suggest that the translation process fails to sufficiently preserve the essential visual cues of the road surface condition. The authors need to discuss the domain adaptation issues between the converted images generated by their method and the real samples.

In summary, the experimental validation section of this paper requires significant improvement, and thus a Major Revision is recommended.

Round 2

Reviewer 1 Report

Comments and Suggestions for Authors

Minor revision comments:

- The phrase “robust classification approach” and “stably classify road conditions” is overstated given that failure cases are explicitly shown in Figures 14–16. Revise to “more robust than time-based switching methods” or “improved robustness under dusk conditions” to align claims with evidence.

- While conceptually clear, the operational definition remains vague. Explicitly state how images were filtered to exclude dusk (Ex: illuminance threshold, solar elevation angle, or manual exclusion). This is essential for reproducibility.

- The argument that newer models “hallucinate textures” is asserted but not empirically supported. Add one citation or a brief qualitative comparison (even anecdotal) or soften the claim to “may risk”.

- The hidden layer size (350 neurons) and λ = 1e−6 appears arbitrary. Add one sentence clarifying whether these were tuned empirically or adopted from prior work.

- Night is defined as 16:00–20:00, which overlaps with dusk in winter but not in other seasons or regions. Add a clarification sentence that this split is geographically and seasonally specific to Hokkaido winter, not universal.

- You claim “no significant difference” between PM and CM2 during dusk, but later argue PM is “practically superior.” Explicitly distinguish statistical significance vs. safety-critical relevance to avoid logical ambiguity.

- The statement that CycleGAN “does not need to be applied to every frame” is reasonable but speculative. Rephrase to “can potentially be applied at a lower frequency” unless experimentally validated.

- When Ice <-> Snow misclassification occurs (Figures 14–16), add one sentence discussing the potential safety implication (false negative vs false positive).

Reviewer 4 Report

Comments and Suggestions for Authors

The author has responded to my comments and made some modifications, so I believe this manuscript is now acceptable.
